# Topological dislocation modes in three-dimensional acoustic topological insulators

Liping Ye[1], Chunyin Qiu [1✉], Meng Xiao [1✉], Tianzi Li[1], Juan Du[1], Manzhu Ke [1✉] & Zhengyou Liu [1,2✉]

Dislocations are ubiquitous in three-dimensional solid-state materials. The interplay of such real space topology with the emergent band topology defined in reciprocal space gives rise to gapless helical modes bound to the line defects. This is known as bulk-dislocation correspondence, in contrast to the conventional bulk-boundary correspondence featuring topological states at boundaries. However, to date rare compelling experimental evidences have been presented for this intriguing topological observable in solid-state systems, owing to the huge challenges in creating controllable dislocations and conclusively identifying topological signals. Here, using a three-dimensional acoustic weak topological insulator with precisely controllable dislocations, we report an unambiguous experimental evidence for the long-desired bulk-dislocation correspondence, through directly measuring the gapless dispersion of the one-dimensional topological dislocation modes. Remarkably, as revealed in our further experiments, the pseudospin-locked dislocation modes can be unidirectionally guided in an arbitrarily-shaped dislocation path. The peculiar topological dislocation transport, expected in a variety of classical wave systems, can provide unprecedented control over wave propagations.

[1] Key Laboratory of Artificial Micro- and Nano-structures of Ministry of Education and School of Physics and Technology, Wuhan University, Wuhan 430072, China. [2] Institute for Advanced Studies, Wuhan University, Wuhan 430072, China. ✉email: cyqiu@whu.edu.cn; phmxiao@whu.edu.cn; mzke@whu.edu.cn; zyliu@whu.edu.cn

Dislocations are rather ubiquitous in three-dimensional (3D) solid-state materials and their existence may significantly modulate the physical properties of the systems[1,2]. As topological line defects in real space, the dislocations are characterized by Burgers vector **B** and cannot be removed by local perturbations due to the conservation of **B**. The dislocations can be classified into two elementary types according to their orientations with respect to **B**: screw dislocations and edge dislocations, whose dislocation lines are parallel and perpendicular to **B**, respectively. In general, a dislocation line can be a combination of these two simple types, which either forms a loop or branches into a network owing to the conservation of **B**.

In recent years, topological matter has emerged as a major branch in broad areas of physics, from condensed matter[3,4] to cold-atom[5,6] and classical systems[7–10]. Intriguingly, the interplay of the dislocation (a real-space topology) with the emergent band topology defined in reciprocal space can induce many fantastic transport phenomena[11–19], such as abnormal conductance and chiral magnetic effects owing to the presence of one-dimensional (1D) gapless topological modes bound to the dislocations. In particular, it has been revealed that the topological dislocation modes (TDMs) can serve as a direct probe to detect the bulk topology of a 3D topological insulator (TI)[11–14]. Specifically, the number of the helical TDMs is given by $\sum_{i=1}^{3} \nu_i \mathbf{b}_i \cdot \mathbf{B}/2\pi$[11–14], where $\nu_i$ and $\mathbf{b}_i$ are the weak topological index[20,21] and primitive reciprocal lattice vector of the TI along the $i$th direction, respectively. According to this bulk-dislocation correspondence, the weak topological index of the 3D TI can be experimentally identified by the TDMs through adjusting the amplitude and orientation of **B**. The intriguing bulk-dislocation correspondence in 3D TIs[11–14], although established soon after the discovery of symmetry-protected quantum phases[3,4], has not been unambiguously confirmed in experiments so far, mostly owing to the shortness of appropriate TI materials, and the huge challenges in creating controllable dislocations and conclusively identifying the TDM signals[18,19]. All the experimental obstacles can be overcome in artificial crystals of classical waves, benefited from their exceptional macroscopic controllability[22–26].

Here, we experimentally construct a 3D acoustic TI (ATI) with precisely-controlled dislocations and present a smoking-gun for the long-sought bulk-dislocation correspondence. The delicate design of our acoustic system enables us to observe not only the TDMs in momentum-resolved frequency spectroscopy but also their spatial localization in pressure-field distributions. The topological robustness of the TDMs is identified by introducing a spin-preserved defect to the dislocation. Significantly, comparing with the topological defect modes recently revealed in two-dimensional (2D) systems[22–25], the TDMs observed here exhibit more flexibilities in wave manipulations since the dislocation lines can be deformed at will in 3D space. As such, we can design dislocation waveguides of arbitrary shapes and unidirectionally guide the TDMs along any prescribed routes inside the bulk materials, which are conclusively identified by our acoustic experiments.

## Results

**TDMs in a 3D TI**. Prior to entering the details of our acoustic system, we present a brief introduction for the 1D TDMs in a 3D TI. We start with 2D quantum spin Hall (QSH) effect. As illustrated in Fig. 1a, a finite QSH insulator can support a pair of spin-locked helical modes at its boundaries, which are scattering immune to any spin-conserved defects such as the sample corners. Then we stack the QSH layers to form a 3D TI and create a $z$-directed screw dislocation inside it by a cut-and-glue procedure sketched in Fig. 1b. (Consider first the screw dislocation for

simplicity.) By intuition, in the limit of weak interlayer couplings, the screw dislocation can support spin-locked 1D gapless modes that spiral up or down along the dislocation line[11,27]. Such TDMs persist in the presence of finite interlayer couplings, as long as the nontrivial band gap of the 3D TI is not closed[11,12]. Physically, the TDMs arise from a delicate interplay between two different Berry phases: one associated with cycles in the Brillouin zone and the other represented by the Burgers vector of the dislocation. Just like the 2D QSH effect, the 1D gapless TDMs are protected by time-reversal symmetry and topologically robust against spin-preserved lattice disorders or defects[11–14]. This is illustrated in Fig. 1c with an additional defect created around the dislocation line, in which the TDMs can bypass the defect smoothly. Rich dislocation paths can be constructed if combining the screw and edge dislocations together. Interestingly, as exemplified in Fig. 1d with a simple dislocation loop, the spin-locked TDMs can be guided along an arbitrary dislocation path without scattering[14]. This extraordinary topological dislocation transport, which has yet to be observed in any systems, will be experimentally demonstrated in our acoustic system.

**Acoustic TDMs and their topological transports**. Our 3D ATI is stacked layer-by-layer with 2D acoustic analog of QSH insulators for airborne sound. As shown in Fig. 2a, each acoustic analog of QSH layer consists of a square lattice of site ring-waveguides (yellow) in the $x$–$y$ plane, which are coupled through straight tubes (blue) and coupler ring-waveguides (gray). All structure parameters are optimized to ensure strong couplings and meanwhile favor the fabrication of a real sample (see details in Supplementary Fig. 1). Although unlike real QSH insulators, there is no Kramers degeneracy for acoustic systems. The strongly coupled 2D waveguides network serves as a good acoustic analog of the QSH insulator[28–31], where the anti-clockwise and clockwise circulations of sound in the site ring-waveguides mimic the (pseudo)spin-up and -down degree of freedoms (defined in the inset of Fig. 2a). The transfer matrix method based theoretical analysis of this 2D insulator can be found in Supplementary Fig. 2. The effectiveness of our 2D design has been fully confirmed by our acoustic experiments (see Supplementary Figs. 3 and 4). To construct a 3D ATI, vertical narrow tubes (purple) are introduced to couple the site ring-waveguides in the $z$ direction. The vertical tubes are printed narrow for fulfilling weak interlayer coupling, such that the nontrivial topology of the 2D QSH system is inherited (see Supplementary Fig. 5), meanwhile, with the careful design of their distribution, to avoid visible pseudospin flip (see Supplementary Fig. 6). As illustrated in Fig. 2b, the bulk topology of the 3D ATI, characterized by the weak topological indices $(\nu_1 \nu_2 \nu_3) = (001)$[20,21], exhibits as topological surface states only on the side surfaces (see details in Supplementary Fig. 7). Note that the fact of lacking topological surface states on the $x$–$y$ surface favors our experimental real-space visualization of the TDM signals bounded to the edge dislocation (or the termination of the screw dislocation) on the top surface of the sample (see Figs. 3c, f and 4b). This is different from the (strong) topological insulators considered in solid-state systems, where the hallmark signature of TDMs mixes with the surface signal unavoidably[11,18,19].

Below, we use 1D TDMs predicted by bulk-dislocation correspondence[11–14], rather than 2D topological surface states dictated by conventional bulk-boundary correspondence[20,21], to reveal the nontrivial bulk topology of our 3D ATI. Consider first a screw dislocation that has a unit Burgers vector in the $z$ direction, $\mathbf{B} = (0, 0, H)$. To construct the dislocation, as illustrated in Fig. 2c, a chain of the straight tubes and coupler ring-waveguides (collectively dubbed as in-plane couplers) is torn open and tilted

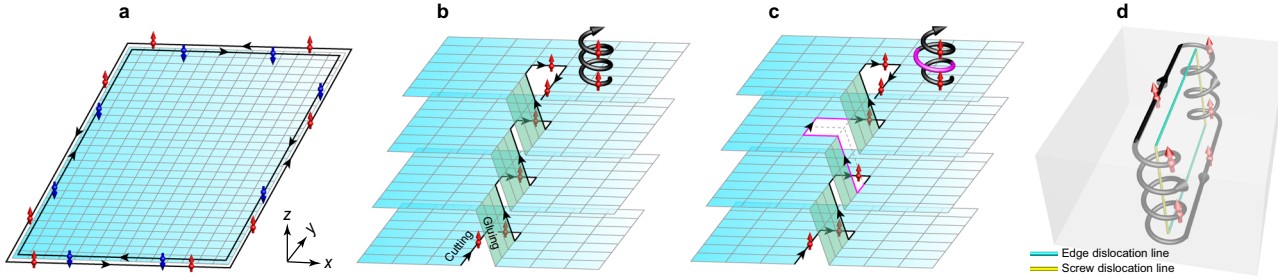

**Fig. 1 TDMs guided by the dislocations in a 3D TI. a** Sketch of 2D QSH effect, featuring one pair of spin-locked edge modes at the boundaries of a finite QSH insulator. The red and blue arrows denote the spin-up and spin-down modes, respectively, and the black arrows indicate their propagating directions. **b** A TDM running spirally along the screw dislocation line in a layer-stacked 3D TI, as visualized vividly with the bold spiral arrow. For brevity, in **b**–**d** we only sketch the spin-up mode and omit its time-reversal counterpart. **c** Scattering-immunity of the TDM to a spin-preserved defect (highlighted with magenta). **d** Unidirectional propagation of the spin-locked TDM in a dislocation loop connected with edge and screw dislocations.

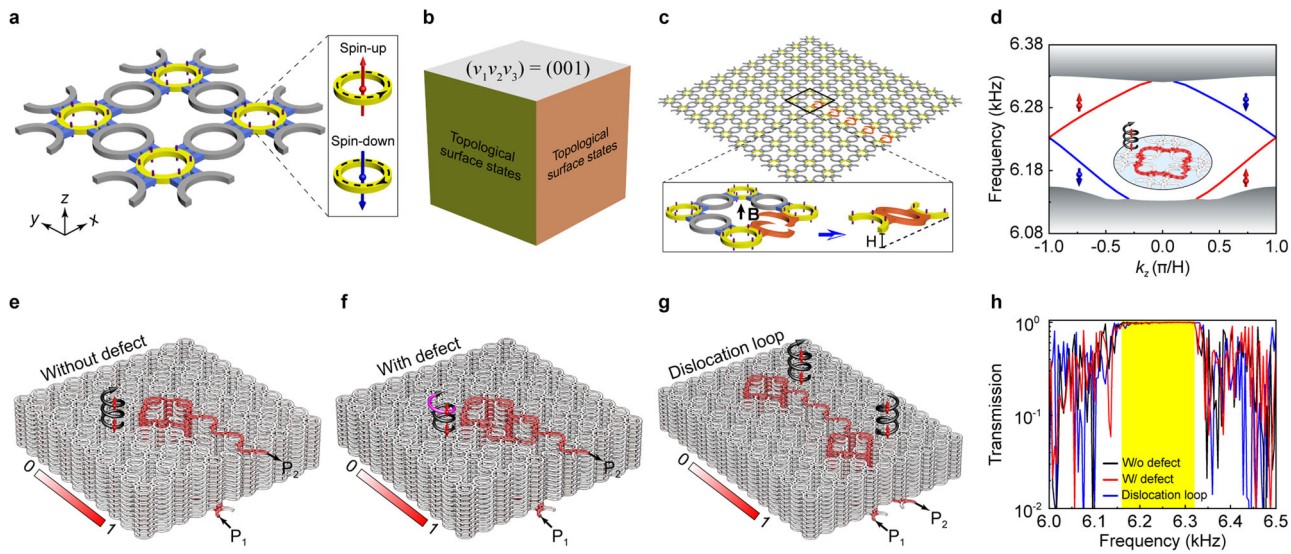

**Fig. 2 Numerical demonstrations of the acoustic TDMs and their topological transports. a** Monolayer structure used for stacking our 3D ATI. It consists of a square lattice of site ring-waveguides (yellow) connecting with straight tubes (blue) and coupler ring-waveguides (gray) in the x–y plane, together with vertical narrow tubes (purple) for interlayer couplings in the z direction. The in-plane and out-of-plane lattice constants are $a = 208$ mm and $H = 27.5$ mm, respectively. The inset defines the pseudospins according to the circulating directions of sound in the site ring-waveguides. **b** Bulk topology of the 3D ATI, manifested as topological surface states only on the side surfaces. **c** Supercell used to simulate the acoustic TDMs. The screw dislocation is created by replacing a chain of the in-plane couplers with titled interlayer couplers (orange). The inset amplifies the dislocation and presents an equivalent but more intuitive demonstration for the tilted interlayer coupler by considering the periodicity of the system along z axis. **d** Dislocation-projected band structure. The color lines represent helical TDMs and the gray shadows represent bulk states. Inset: Pressure amplitude distribution exemplified for a pseudospin-up TDM at 6.27 kHz, where the bold spiral arrow visualizes its propagation. **e** Field pattern simulated at 6.27 kHz for a finite sample that contains a screw dislocation in its interior and two edge dislocations on the top and bottom layers. **f** The same as **e** but for the system with a defect on the top layer. **g** Similar simulation for a sample with a dislocation loop. **h** Transmission spectra for the three dislocation systems in (**e**–**g**), which characterize quantitatively the topological robustness of the TDMs within the nontrivial band gap (yellow shadow).

to connect the neighboring layers. Again, the structure of the tilted interlayer coupler is optimized to ensure a strong coupling over the interested frequency range (see Supplementary Fig. 8). Figure 2d gives the dislocation-projected band structure (see "Methods"). Clearly, it shows a pair of counter-propagating TDMs across the bulk gap, as a direct reflection of the nontrivial bulk topology of our 3D ATI. The gapless TDMs are pseudospin-momentum locked and strongly localized at the dislocation, as exemplified by a pseudospin-up mode in the inset. Without data provided here, gapless TDMs hosted by the edge dislocation are demonstrated in Supplementary Fig. 9. Moreover, we have designed a screw dislocation with $\mathbf{B} = (0, 0, 2H)$, which traps two pairs of 1D helical TDMs, as the further numerical evidence for the bulk-dislocation correspondence (see Supplementary Fig. 10).

To elucidate the transport properties of the acoustic TDMs, we consider first a 10-layer sample (Fig. 2e) that hosts a screw dislocation in its interior, accompanying with one edge dislocation on the top layer and another in the bottom (see structure details in Supplementary Fig. 11). As shown in Fig. 2e, the pseudospin-up TDM, selectively excited by the sound source locating at the inlet $P_1$, propagates first along the edge dislocation in the bottom layer, then travels spirally up along the screw dislocation, and finally exports from the outlet $P_2$ of the edge dislocation on the top layer. It is of interest that there is no visible change exhibited in the amplitude of the sound signal during its propagation, even suffering the sharp transition between the screw and edge dislocations, because of the negligible pseudospin-flip in our coupled ring-waveguides system. The scattering immunity persists in the system with

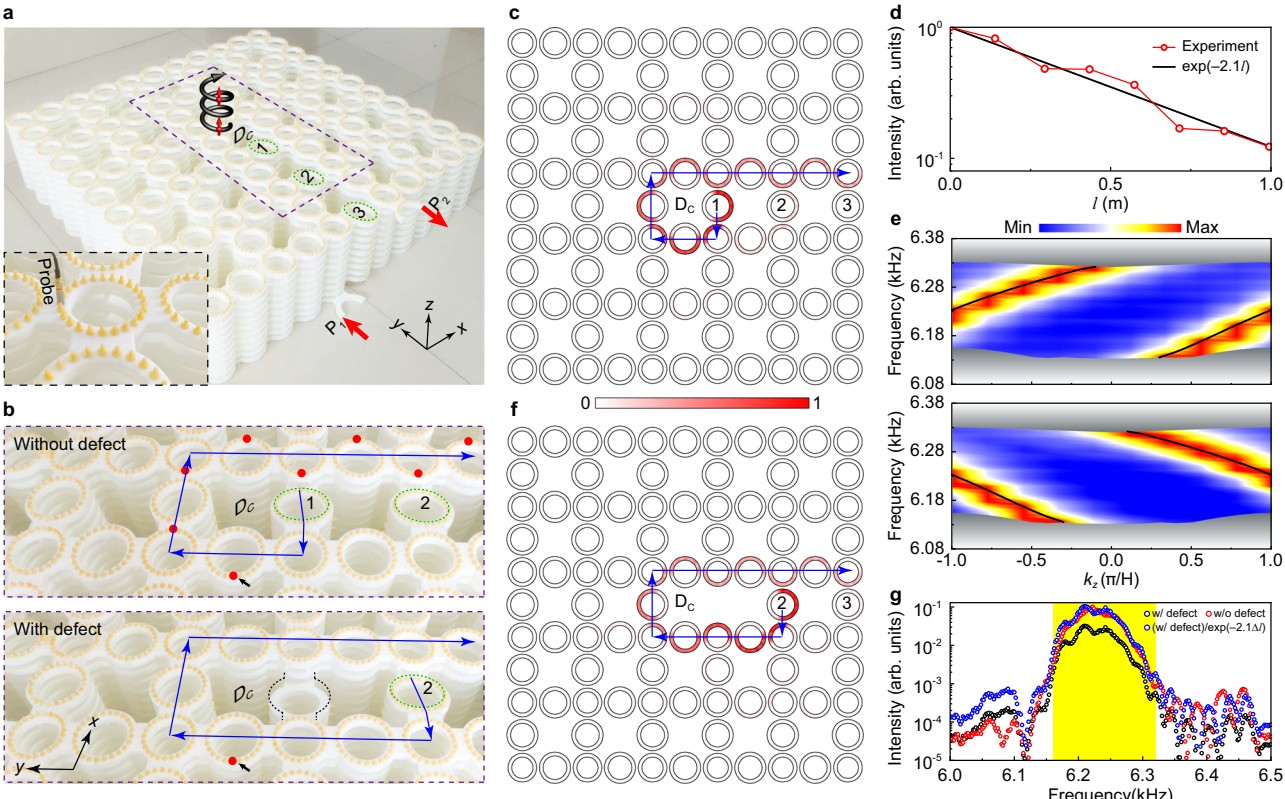

**Fig. 3 Experimental characterizations of the acoustic TDMs and their topological robustness to defects. a** Experimental setup. For clarity, three tilted ring-waveguides on the top layer are numbered and the termination of the screw dislocation is labeled with $D_C$. The ports $P_1$ and $P_2$ are inlet and outlet of sound, respectively. The inset amplifies the details of the top layer, on which subwavelength holes are perforated for inserting the sound probe. **b** Top panel: Zoom-in photograph for the area marked by the purple dashed box in (**a**), where the blue arrows indicate the propagations of the TDMs after leaving the screw dislocation. Bottom panel: The same as the top panel, but with a defect (outlined by the black dashed lines) created near $D_C$. **c** Pressure amplitude pattern scanned on the top layer at 6.27 kHz, which visualizes the presence of TDMs. **d** Sound intensity (red circles) measured at 6.27 kHz for a sequence of sites along the dislocation path (red dots in the top panel of **b**). The black line shows an exponential fit, where $l$ is the propagation length measured from the first site, i.e., the red dot specified with a black arrow. **e** Top panel: Measured dispersion for the 1D TDMs (bright color), which captures precisely the simulation result (black lines). Bottom panel: The same as the top panel, but the sound is injected from the port $P_2$ and leaves from the port $P_1$. **f** The same as **c**, but for the system with a defect. **g** Sound intensity spectra detected at the sites specified by black arrows in **b** for the systems with and without a defect. $\triangle l = 0.56$ m is the extra propagation length of the TDMs induced by the defect.

pseudospin-preserved defects, as exemplified in Fig. 2f, in which one tilted interlayer coupler around the screw dislocation is removed (see details in Supplementary Fig. 11). (For the convenience of demonstration, the defect is created on the top layer of the sample.) Based on a similar reason, one may conclude that an injected sound signal will experience no scattering when propagating in an arbitrarily-shaped dislocation path. Here we consider a simple dislocation loop as sketched in Fig. 1d, which consists of two screw dislocations and two edge dislocations connected head-to-tail alternatively. Again, for clarity the edge dislocations locate at the top and bottom layers (see details in Supplementary Fig. 11). Figure 2g shows the simulation result, which is in accordance with our expectation. To evaluate the topological robustness of the TDMs more quantitatively, in Fig. 2h we present power transmission spectra for the above three dislocation samples. It shows that for all the cases nearly perfect transmissions occur within the nontrivial band gap, while suffering striking reflections beyond that frequency range. Such pseudospin-locked unidirectional sound propagation is further demonstrated by a sample with a much more complex dislocation path (see Supplementary Fig. 12). The exceptional performance of sound routing in 3D space will be particularly useful for designing new conceptional acoustic devices.

**Experimental observations of the acoustic TDMs and their topological transports.** The presence of the 1D TDMs has been directly visualized in our airborne sound experiments. Figure 3a shows the experimental sample fabricated precisely with 3D printing. It has a geometry exactly the same as that used in Fig. 2e, provided with one screw dislocation in its interior and two edge dislocations on the top and bottom surfaces. To detect desired sound signals, subwavelength holes are perforated in some ring-wave guides, which are sealed when not in use. As considered in Fig. 2e, a point-like sound source is positioned at the port $P_1$ in the bottom layer, and a sound probe is inserted into the small holes to detect the pressure distribution (see the inset in Fig. 3a). In this setup, the pseudospin-up TDMs will be selectively excited, which spiral up along the screw dislocation and leave from the top-layer edge dislocation as sketched in Fig. 3b (top panel). This process can be visualized by scanning the pressure profile on the top layer. Figure 3c presents the experimental data at 6.27 kHz. It shows that the sound signal indeed emerges from the tilted ring-waveguide 1 and turns round to the prescribed edge dislocation, along which the pressure field is strongly localized. The experiment captures well the simulation result in Fig. 2e, except for the sound attenuation due to the presence of dissipation in real experiments. To evaluate the energy dissipation more quantitatively, in Fig. 3d we have fitted the sound intensity

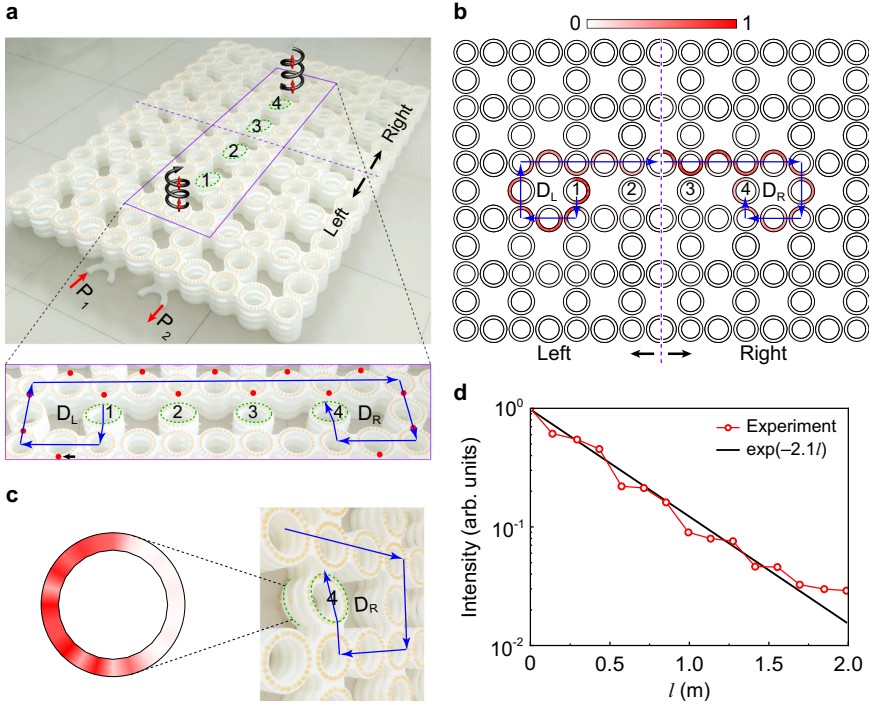

**Fig. 4 Experimental observation of the unidirectional sound routing in a dislocation loop. a** Experimental sample. The inset gives more details around the dislocations, where $D_L$ and $D_R$ label the positions of the screw dislocations, and the blue arrows point to the propagation of the pseudospin-up TDMs excited by the sound source at the port P₁. **b** Pressure amplitude distribution scanned on the top layer at 6.27 kHz. **c** Pressure amplitude distribution measured for the lowest ring-waveguide around the screw dislocation $D_R$. **d** Sound intensity (red circles) extracted along the dislocation path on the top layer, whose shape is captured well again by $\exp(-2.1l)$ (black line).

distribution at 6.27 kHz along the dislocation path, which gives an estimation of the energy dissipation coefficient 2.1 m⁻¹. Note that this setup also allows an experimental characterization of the strong coupling induced by the tilted interlayer couplers (see Supplementary Fig. 13), which is a key factor that determines the effectiveness of the 3D dislocation sample, in addition to the strong in-plane coupling that has already been examined in the 2D QSH system (see Supplementary Fig. 3).

To present a conclusive experimental evidence for the bulk-dislocation correspondence, we have further measured the pressure field along the screw dislocation line and mapped out the TDM dispersion by performing 1D Fourier transformation (see Methods). As shown in the top panel of Fig. 3e, the experimentally measured TDM dispersion (bright color) exhibits clearly a gapless 1D mode with a positive slope. It reproduces perfectly the simulated dispersion (black line) for the pseudospin-up TDMs that move up along the screw dislocation, except for the band broadening due to the finite-size effect. Similarly, we have also identified the pseudospin-down TDMs by injecting sound waves from the port P₂. The measured data is provided in the bottom panel of Fig. 3e, which exhibits a negative slope as expected. The presence of one pair of gapless helical TDMs reflects faithfully the nontrivial bulk topology of our 3D ATI, which in turn supports the idea of using the dislocation as a valuable bulk probe[11–14]. Moreover, the experimentally measured dispersions confirm the prediction (Fig. 2e, h) that sound can cross the sharp transition smoothly (without scattering) from the screw dislocation to the edge dislocation, since there is no signal of the pseudospin-down (pseudospin-up) TDMs in the top (bottom) panel of Fig. 3e.

Now we turn to confirm the topological robustness of the TDMs against defects. The experimental sample has a geometry employed in Fig. 2f. As shown more clearly in the bottom panel of Fig. 3b, the defect is constructed by removing one tilted

interlayer coupler adjacent to the dislocation, i.e., the tilted ring-waveguide 1 and the associated straight tubes. In this case, the TDMs will be coupled to the top layer through the tilted ring-waveguide 2. To confirm this point, we have experimentally scanned the pressure field on the top layer of the sample. Figure 3f exemplifies the data at 6.27 kHz. As predicted in Fig. 2f, the sound field bypasses the defect and propagates along the prescribed path like the system without the defect. To quantitatively characterize the topological robustness of the TDMs against the pseudospin-preserved defect, we have further extracted and compared the sound energy spectra at the positions highlighted in Fig. 3b (the red dots with black arrows) for the systems without and with a defect. As shown in Fig. 3g, the sound intensity spectra of the two systems (black and red circles) deviate with each other over the entire frequency range. However, if the dissipation induced by the extra propagation length of the TDMs is compensated, the spectrum of the defect system (blue circles) inside the nontrivial gap (highlighted in yellow) exhibits almost the same magnitude with that of the perfect system (red circles).

To identify the capability of the unidirectional sound routing in a prescribed dislocation path, we consider a 4-layer sample with a dislocation loop like Fig. 2g. As shown in Fig. 4a, the two screw dislocations labeled with $D_L$ and $D_R$ are connected by a pair of edge dislocations on the top and bottom layers. The sound source is located at the port P₁ in the bottom layer, which excites selectively the pseudospin-up TDMs in the dislocation loop. In Fig. 4b we present the sound field scanned on the top layer. It shows that the sound signal indeed travels up along the screw dislocation $D_L$ and propagates along the edge dislocation on the top layer. (Note that the sound signal attenuates as its propagation due to the inevitable absorption. For the clarity of demonstration, the field region is divided into two parts and the data of each area is normalized by the corresponding maximum value.) To verify the following downward propagation along the

screw dislocation $D_R$, we have detected the sound signal inside the tilted ring-waveguide at the bottom layer, as shown in Fig. 4c. The field distribution (exemplified at 6.27 kHz) exhibits its right pattern expected for the pseudospin-up TDM. The pseudospin-locked unidirectional sound propagation can be further seen in the sound intensity distribution extracted along the dislocation path (Fig. 4d), which gives the same estimation of energy dissipation coefficient as the systems in Fig. 3. (The consistence exhibited in broadband spectra can be seen in Supplementary Fig. 14). This dissipation signature, together with the above field distributions, confirms the physical picture introduced in Fig. 1d: the TDMs can be guided in any prescribed dislocation paths without scattering.

## Discussion

In summary, we have presented an unambiguous observation of the long-desired TDMs in a 3D ATI via real-space visualization and momentun-space spectroscopy. Our experiments identify the nature of the dislocation as a probe in revealing the bulk topology of 3D TIs[11–14]. Dramatically, we have demonstrated that the pseudospin-locked TDMs can be guided without scattering along an arbitrary dislocation path in 3D space. The capability of freely unidirectional routing waves in 3D is far beyond that exhibited in the early reported artificial crystal-based waveguides[32,33] or the newly developed topological edge/surface/hinge states-based topological waveguides[34–46]. Concretely, for example, compared with topological surface states in 2D which exist only on the boundaries or interfaces, and the 1D topological hinge states propagating along the usually straight hinges of 3D topological crystals[41–46], the TDMs can propagate unidirectionally along the dislocations with various shapes created anywhere inside the bulk or at the surface of 3D topological crystals[11–14]. The dissipation loss (common for most waveguide systems) exhibited here will be a destructive issue in practical applications. However, for some cases, such as photonic systems composed of pure dielectrics, this problem will be greatly reduced. On the theoretical side, our findings will stimulate the study on the highly-intriguing interaction between the real-space topology and band topology. On the application side, the peculiar topological dislocation transport points to new possibilities for information communication and energy transportation.

## Methods

**Simulations**. All numerical simulations are performed by COMSOL Multiphysics, a commercial finite-element solver package. The resin used for sample fabrication is modeled as acoustically rigid for the airborne sound with speed 344 m/s. To simulate the TDMs in Fig. 2d, we consider a supercell of $10 \times 10$ lattice sites in the $x$–$y$ plane, in which 5 in-plane couplers are replaced by the tilted interlayer couplers (Fig. 2c). Besides the Bloch boundary condition applied along the $z$ direction, periodic boundary conditions are used in the $x$ and $y$ directions. As such, a pair of separated screw dislocations with opposite Burgers vectors is created simultaneously, one at the center and the other at the boundary, respectively. For brevity, in Fig. 2d we selectively plot the dispersion of the TDMs trapped at the center of the supercell, which can be distinguished according to their field distributions. The pressure amplitude distributions in Fig. 2e–g are simulated for three finite-sized samples. The transmission in Fig. 2h is defined as the ratio between the sound-energy flows extracted at the ports $P_1$ and $P_2$.

**Experiments**. Our experimental samples are fabricated by 3D printing with a fabrication error of ~0.1 mm. Circular holes of radii ~3.5 mm are perforated on specific ring-waveguides for inserting the sound probe. To obtain the pressure field distributions in Fig. 3c, f, and b, sound waves are launched from a subwavelength tube of radius ~3.5 mm, and detected hole-by-hole through a 1/4 inch microphone (B&K Type 4958), together with another identical microphone fixed for phase reference. The amplitude and phase of the pressure fields are recorded and frequency-resolved by a multi-analyzer system (B&K Type 3560B). In order to attain the dispersion of the TDMs (Fig. 3e), 1D Fourier transformation is performed for the pressure field extracted along the dislocation line. The sound energy distribution in Fig. 3d or Fig. 4d is normalized by the corresponding maximum value.

## Data availability

The data that support the findings of this study can be obtained from the Zenodo database at https://doi.org/10.5281/zenodo.5802548.

## Code availability

Numerical simulations in this work are all performed using the acoustic module of a commercial finite-element simulation software (COMSOL MULTIPHYSICS). All related codes can be built with the instructions in the "Method" section. The codes for numerical simulations are also available in the Zenodo database at https://doi.org/10.5281/zenodo.5802548.

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

## Acknowledgements

This work was supported by the National Natural Science Foundation of China (grant numbers 11890701, 11904264, 11774275, 11974262, 12047542, and12104347), the Young Top-Notch Talent for Ten Thousand Talent Program (2019–2022), the Fundamental Research Funds for the Central Universities (grant number 2042020kf0209), the National Key R&D Program of China (grant number 2018YFA0305800), and the China Post-doctoral Science Foundation (grant numbers 2020TQ0233, 2020M682462).

## Author contributions

C.Q., M.X., and Z.L. initiated and supervised the project. L.Y. did the simulations and designed the samples. L.Y., T.L., J.D., and M.K. performed the experiments. L.Y. wrote the draft. C.Q., M.X., and Z.L. analyzed the data and revised the manuscript. All authors contributed to scientific discussions of the manuscript.

## Competing interests

The authors declare no competing interests.
