## [Peer review file · Nature Communications]

REVIEWER COMMENTS

Reviewer #1 (Remarks to the Author):

This is a very impressive work, considering the complexity of the design and volume of the experimental work and simulations. I am generally positive for the main text as well. However, there are two issues that could be critical and un-answered in the manuscript:

1. Strictly speaking in physics, there is no quantum spin Hall effect for acoustic waves (in the strict topological sense), since there is no Kramers degeneracy for phonons. I understand that the authors are designing the system to mimic such an effect. However, this fact should be exposed to some extent so that the audience is not misled.
2. With such knowledge there is no strict weak topological insulator, as shown in the literature (Nat. Commun. 9, 4555 (2018); Phys. Rev. Lett. 123, 195503 (2019); Nat. Commun. 11, 2318 (2020)), acoustic weak topological insulators all have some issues such as having a mini gap in the edge states spectrum or the need for additional symmetries such as the crystalline symmetries. If there is a mechanism in the system studied here, what is the mechanism making the edge states more robust?
3. In the above sense, the authors should say clearly that they are establishing an analog in acoustic system in somewhere in the manuscript. That is, this effect should be present cautiously to avoid being misleading to the society.
4. There are solid experiments and simulations, while the theory part is obviously lacking. I cannot even find the Hamiltonian of the topological insulator in the supplementary information.

Overall, the manuscript needs to be considerably revised before publication.

Reviewer #2 (Remarks to the Author):

In this manuscript, the authors experimentally demonstrated topological dislocation states in 3D acoustic crystals. Based on stacking a 2D network of coupled acoustic resonators into 3D with weak interlayer coupling, 3D acoustic crystals are constructed with topological behaviors inherited from the native 2D system. By introducing screw dislocation with some strong tilted interlayer coupling, a pair of gapless 1D topological dislocation states are observed. Generally, the results are reasonable and correct. The manuscript is clearly written and well organized. I am happy to recommend this manuscript publication in Nature Communications if the authors can well address the following issues.

1. It may be true that the weak indices of the current model can be written as $(0;0,0,1)$, but such direct stacking of the 2D quantum spin Hall system with extremely weak (even without) interlayer coupling cannot be treated as a real "3D" topological insulator. There is nearly no dispersion along the k_z -direction as shown in Fig. S5. Compared to previous studies [e.g., Nat. Commun. 11, 2318(2020) & PRL 123, 195503(2019)], there are no Dirac-like surface states in this 3D model. Although the geometry dimension is 3D, in my opinion, a more accurate description should be "multilayer quantum spin Hall system" or "3D topological acoustic crystal" to avoid misunderstanding. This issue needs to be discussed.
2. In the manuscript, two pairs of gapless 1D topological dislocation states are realized with a screw dislocation $\mathbf{B} = (0,0,2H)$, see Fig. S8. However, according to Ref. 14 [PRB 90, 241403(R) (2014)], "In contrast, in a weak phase characterized by the weak topological index M , protected helical modes are obtained only when the product $M \cdot b \pmod{2\pi}$ is nontrivial (page 3)." Does it mean these two pair states may be coupled to form a gap even without pseudospin flipping?
3. The tilted interlayer coupler is a key factor to create a screw dislocation, which is optimized to ensure a strong coupling in this work. The tilted degrees β is chosen to be 15 In Fig. S6 and 27.6 in Fig. S8. Would such tilted coupling (or larger β) make pseudospin flipping? Does there exist topological phase transition between strong and weak tilted interlayer couplings with the different β ?
4. It would be better if the authors can compare their 1D topological dislocation states to the other 1D

topological hinge states in 3D topological crystals.

5. In Fig. 3d and Fig. 4d, the authors estimate the sound dissipation coefficient is $2.1(1/m)$ along the dislocation path. How about the dissipation coefficient in the air at the same frequency?

Reviewer #3 (Remarks to the Author):

This manuscript presents a theoretical and experimental study of topological defect modes (TDMs) and bulk-dislocation correspondence in weak 3D acoustic topological insulators. The system under study consists of a 3D topological insulator stacked by layers of 2D quantum spin Hall insulators with weak inter-layer couplings. A screw dislocation is added to the stacking direction to adjust the Burgers vectors, and pairs of TDMs with spin-momentum locking are supported in the bulk gap and are spatially localized at the dislocation. The number of TDM pairs is determined by the Burgers vectors, and allows to probe the bulk topology, in this case, the weak topological indices. Experimentally, the authors study the acoustic waves coupled to the TDMs propagating along the dislocation, measure the dispersion and transmittance of the TDMs, and demonstrate the robustness of the TDMs against spin-conversing perturbations.

Overall, the presented results are new and interesting, and the manuscript is well written and concise. The observation of TDMs and bulk-dislocation correspondence in 3D acoustic systems is timely and may stimulate efforts in other classical wave systems. The manuscript should be of interest to the topological physics community. I thus recommend the manuscript to be accepted in Nature Communications. I also have a few comments regarding the details of the manuscript for the authors to consider.

1. The manuscript currently is focused on weak 3D TIs, and this is worth mentioning in the title and abstract.
2. In the examples studied here, the screw dislocations are all embedded within the bulk of the system to allow unambiguous observation of TDMs. I wonder how the TDMs interact with the QSH surface states, if, say, the dislocation is added at the xz or yz planes.
3. The interlayer couplings are designed to be weak in the current results. I wonder how the TDMs depend on the interlayer couplings? How does the result change, if the xy plane also supports surface states, i.e., for a strong 3D TI?
4. The authors claim that the topological dislocation transport provides unprecedented control over wave propagations and new possibilities for applications, but do not expand further on the application side. It would be helpful if these claims can be expanded. And comparisons on topological surface states in 2D and TDMs in 3D could also be helpful, since unidirectional wave routing is also achievable in 2D systems.

Responses to Referees

Summary of comments and responses:

We have received three reports. All the referees are very positive on our work. The first referee commented that *“This is a very impressive work, considering the complexity of the design and volume of the experimental work and simulations. I am generally positive for the main text as well.”* The second and third referees have recommended the publication of our manuscript in Nature Communications. The second referee pointed out *“the results are reasonable and correct. The manuscript is clearly written and well organized. I am happy to recommend this manuscript publication in Nature Communications...”* and the third referee pointed out *“Overall, the presented results are new and interesting, and the manuscript is well written and concise.”* and *“The manuscript should be of interest to the topological physics community. I thus recommend the manuscript to be accepted in Nature Communications.”*

Below, we provide our detailed replies to all the questions and suggestions of the referees, which are insightful and constructive, and improve our manuscript. Our manuscript has been carefully revised accordingly (**highlighted in yellow**). We think our current version should be suitable for publication in Nature Communications. Thank you very much for your reconsideration.

Reviewer #1

This is a very impressive work, considering the complexity of the design and volume of the experimental work and simulations. I am generally positive for the main text as well. However, there are two issues that could be critical and un-answered in the manuscript:

Reply: We thank the referee for the great efforts in reviewing our paper. We also appreciate the referee for the positive comment “*This is a very impressive work, considering...*”. All the suggestions and comments below are very valuable and greatly helpful to improve our manuscript.

1. Strictly speaking in physics, there is no quantum spin Hall effect for acoustic waves (in the strict topological sense), since there is no Kramers degeneracy for phonons. I understand that the authors are designing the system to mimic such an effect. However, this fact should be exposed to some extent so that the audience is not misled.

Reply: We thank the referee for pointing this out. We agree with the referee’s point that there is no Kramer’s degeneracy for phonons and we are designing an acoustic system to mimic the quantum spin Hall (QSH) effect. In fact, our 3D ATI is a layer-by-layer stacking of 2D strongly coupled waveguides which are proved to be good acoustic analog of QSH insulator (see Refs. [28-31] in the main text).

This point has now been strengthened in the revised manuscript (page 5) “Our 3D ATI is stacked layer-by-layer with 2D acoustic analog of QSH insulators for airborne sound. As shown in Fig. 2a, each acoustic analog of QSH layer consists of a square lattice of site ring-waveguides...” We also point out (page 5) “Although unlike real QSH insulators, there is no Kramers degeneracy for acoustic systems. The strongly coupled 2D waveguides network serves as a good acoustic analog of the QSH insulator²⁸⁻³¹, where the anti-clockwise and clockwise circulations of sound in the site ring-waveguides mimic the (pseudo)spin-up and -down degree of freedoms (defined in the inset of Fig. 2a).”.

We have also modified a few sentences in the revised supplementary information (page 2) “**Supplementary figure 1 Construction of the 2D acoustic analog of QSH insulator.**”, “c, Bulk band structure of the acoustic analog of QSH insulator.”, (page 5) “**Supplementary figure 3 Experimental characterizations for our 2D acoustic analog of QSH insulator.**”, and (page 6) “**Supplementary figure 4 Testing topological robustness of the helical edge modes in the 2D acoustic analog of QSH insulator.**”.

2. With such knowledge there is no strict weak topological insulator, as shown in the literature (Nat. Commun. 9, 4555 (2018); Phys. Rev. Lett. 123, 195503 (2019); Nat. Commun. 11, 2318 (2020)), acoustic weak topological insulators all have some issues such as having a mini gap in the edge states spectrum or the need for additional symmetries such as the crystalline symmetries. If there is a mechanism in the system studied here, what is the mechanism making the edge states more robust?

Reply: We thank the referee for this comment. The referee is correct that the mini gap exists when the dispersions of the surface states cross at the time reversal invariant point in the acoustic weak topological insulators if there is (pseudo)spin mixture. In fact, there is no mini gap of the surface states observed for our current system. The reason is that the surface states inside the nontrivial band gap in our system for two pseudospins are far apart and the dispersion crossing point, if exists, is buried under the projection of bulk bands. Thus, no mini gap can be seen.

3. In the above sense, the authors should say clearly that they are establishing an analog in acoustic system in somewhere in the manuscript. That is, this effect should be present cautiously to avoid being misleading to the society.

Reply: We thank the referee for this constructive suggestion. As outlined in the reply to the first question, now we have rewritten a few sentences in both the main text and supplementary information to accurately describe our system and to avoid misunderstanding.

4. There are solid experiments and simulations, while the theory part is obviously lacking. I cannot even find the Hamiltonian of the topological insulator in the supplementary information.

Reply: We thank the referee for raising this point. We agree that some theoretical discussions can improve the manuscript. As shown in Ref. 29 [Phys. Rev. Lett. 110, 203904 (2013)], the Hamiltonian governing the dynamics of the 2D system cannot be written simply or explicitly, as the strong couplings involved make the tight-binding description inapplicable. Meanwhile, local Hamiltonians, like the $\mathbf{k}\cdot\mathbf{p}$ ones, cannot capture the topology of a whole band. However, one can theoretically investigate such a 2D system with the transfer matrix method as proposed in Ref. [29]. As proposed in Refs. [28-31], the strongly coupled 2D waveguide system in our work is a good acoustic analog of the quantum spin Hall (QSH) insulator.

Liang Fu's work, i.e., Ref. 21 [Phys. Rev. B 76, 045302 (2007)] shows that the weak topological insulator can be interpreted as layered 2D QSH insulators with weak

enough interlayer coupling. Thus, our 3D system stacked by layers of the acoustic analog of the QSH insulator with weak coupling can be safely regarded as acoustic weak topological insulators. Similarly, there is no simple Hamiltonian for the 3D topological insulators either. Due to the complexity of the 3D system, we do not intend to give the transfer matrix based theoretical description, but to respond this point, we have added a new section in supplementary information (pages 3-4) to outline the basic results of the transfer matrix method for 2D insulator. We have also added one sentence in the main text to address this point (page 5) “The transfer matrix method based theoretical analysis of this 2D insulator can be found in supplementary Fig. 2.”

Overall, the manuscript needs to be considerably revised before publication.

Reply: We thank the referee for the positive comments on this work and we hope our above replies have successfully addressed all your concerns. Based on the constructive suggestions from you and the other referees, our revised manuscript has been considerably improved and hopefully the current version is suitable for publication.

Reviewer #2

In this manuscript, the authors experimentally demonstrated topological dislocation states in 3D acoustic crystals. Based on stacking a 2D network of coupled acoustic resonators into 3D with weak interlayer coupling, 3D acoustic crystals are constructed with topological behaviors inherited from the native 2D system. By introducing screw dislocation with some strong tilted interlayer coupling, a pair of gapless 1D topological dislocation states are observed. Generally, the results are reasonable and correct. The manuscript is clearly written and well organized. I am happy to recommend this manuscript publication in Nature Communications if the authors can well address the following issues.

Reply: We thank the referee for the kind recommendation for publication in Nature Communications.

1. It may be true that the weak indices of the current model can be written as $(0;0,0,1)$, but such direct stacking of the 2D quantum spin Hall system with extremely weak (even without) interlayer coupling cannot be treated as a real “3D” topological insulator. There is nearly no dispersion along the k_z -direction as shown in Fig. S5. Compared to previous studies [e.g., Nat. Commun. 11, 2318(2020) & PRL 123, 195503(2019)], there are no Dirac-like surface states in this 3D model. Although the geometry dimension is 3D, in my opinion, a more accurate description should be “multilayer quantum spin Hall system” or “3D topological acoustic crystal” to avoid misunderstanding. This issue needs to be discussed.

Reply: We thank the referee for raising this point. We agree with the referee that the interlayer coupling is kept small, such that our 3D system can inherit the nontrivial topology from the 2D system. However, as we can see from supplementary Figs. 6b and 6d of the revised supplementary information, the band structure of our 3D system does disperse along the k_z -direction, otherwise the projected band will be discrete lines instead of strips.

In general, a 3D topological insulator exhibits Dirac-like surface states at time reversal invariant points in the surface Brillouin zone (at the center or boundary) provided that the surface states cross inside the band gap. Indeed, there are no Dirac-like surface states for our current system. The reason is that the surface states inside the nontrivial band gap in our system for two pseudospins are far apart and the cross of their dispersions is buried under the projection of bulk bands. Dirac cone of surface states may appear if the cross is tuned into the band gap. To illustrate this idea, we consider a similar 2D acoustic topological insulators [proposed in Appl. Phys. Lett.

110, 173505 (2017)], as shown in the right panel of Fig. R1. Compared with our current system, such a system considers a larger site ring-waveguide, and hence the strong coupling region can be optimized to cover a few nontrivial band gaps. The edge band structure is shown in the left panel of Fig. R1, where the red and blue dots represent the edge states with pseudospin-up and pseudospin-down, respectively. The edge state dispersion in our case is similar as those in the band gaps highlighted in yellow, where the edge dispersion crossing points are buried under the projection of bulk bands. There are other nontrivial band gaps, see the one highlighted in green, wherein the edge states of two pseudospins cross along the k_x direction. If one further introduces interlayer coupling by stacking this 2D system along the z direction, there the crossing between two pseudospins will extend to a surface Dirac cone since the dispersion along the k_z direction should also be linear (with no pseudospin scattering).

Meanwhile, Liang Fu's work, i.e., Ref. 21 [Phys. Rev. B 76, 045302 (2007)] shows that the weak topological insulator can be interpreted as layered 2D quantum spin Hall insulators with weak enough interlayer coupling. Thus, we think our 3D system can be safely called acoustic weak topological insulators.

Figure R1: Left panel: Edge dispersion for the system proposed in reference [Appl. Phys. Lett. 110, 173505 (2017)]. The unit for calculating the edge dispersion is shown in the right panel, which is periodic in the x direction and finite in the y direction. For clarity, here we provide only the pseudospin-momentum locked edge modes localized at the lower boundary, where the black dots are the bulk states, the red and blue dots are the edge states with pseudospin-up and pseudospin-down. Right panel: Eigenfield distribution exemplified at 5.9 kHz, which is the same for both pseudospins since they are related by time-reversal symmetry.

2. In the manuscript, two pairs of gapless 1D topological dislocation states are realized with a screw dislocation $\mathbf{b} = (0,0,2H)$, see Fig. S8. However, according to Ref. 14 [PRB 90, 241403(R) (2014)], “In contrast, in a weak phase characterized by the weak topological index M , protected helical modes are obtained only when the product $M \cdot \mathbf{b} \pmod{2\pi}$ is nontrivial (page 3).” Does it mean these two pair states may be coupled to form a gap even without pseudospin flipping?

Reply: We thank the referee for this important question. It is true that for a 3D topological insulator characterized by $v_0 \neq 0, \mathbf{M}_v$ ($2\mathbf{M}_v = \mathbf{G}_v = \sum_{i=1}^3 v_i \mathbf{b}_i$), the protected topological dislocation modes are obtained only when the product $\mathbf{M}_v \cdot \mathbf{B} \pmod{2\pi}$ is nontrivial, as proposed in Refs. [11-14] in the main text. Here the appearance of “mod 2π ” is originated from the fact that the topological insulator belongs to \mathbb{Z}_2 classification. Hence if there are two dislocation modes, these two dislocation modes will couple to form a gap even without spin flipping.

However, the situation is different for our case. As proposed in Refs. [28-31] in the main text, the topological properties of our 2D acoustic topological insulator can be described by the non-zero v_1 ($v_1 = \frac{1}{2\pi} \int_{-\pi}^{\pi} dk \text{Tr}[S(k)^{-1} i\partial_k S(k)]$, where $S(k)$ is the unitary scattering matrix between the neighboring site ring-waveguides) invariant and the number of edge states is given by v_1 . Different from the electronic quantum spin Hall system, here v_1 can be any integer number, i.e., $v_1 \in \mathbb{Z}$. According to Ref. 12 [Phys. Rev. B 82, 115120 (2010)], the number of topological dislocation states is given by $\mathbf{G}_v \cdot \mathbf{B}/2\pi$ (note “mod 2π ” is gone now), similar to the dislocation in a 3D integer quantum Hall insulator according to Ref. 12 [Phys. Rev. B 82, 115120 (2010)]. Thus, the two pairs of dislocation states of the system with $\mathbf{B} = (0,0,2H)$ should be nontrivial and will not be gapped without pseudospin flipping.

To address this point, we have now added one sentence in the caption of supplementary Fig. 9. (page 11) “Note here, unlike the electronic quantum spin Hall system, these two helical TDMs cannot be gapped out with spin preserved interaction”

3. The tilted interlayer coupler is a key factor to create a screw dislocation, which is optimized to ensure a strong coupling in this work. The tilted degrees beta is chosen to be 15 In Fig. S6 and 27.6 in Fig. S8. Would such tilted coupling (or lager beta) make pseudospin flipping? Does there exist topological phase transition between strong and weak tilted interlayer couplings with the different beta?

Q1: Would such tilted coupling (or lager beta) make pseudospin flipping?

Reply: Tilted interlayer coupler has been optimized to ensure a strong coupling within the frequency of interests in this work. As shown in supplementary Fig. 7 of the revised supplementary information, we can see that the power transmission I_2/I_1 of the same pseudospin is larger than 0.85 over the nontrivial gap (yellow shadow), which indicates the pseudospin flipping (if exists) is negligibly small. If the tilted interlayer coupler is not constructed with the optimized geometric parameters, the pseudospin flipping will not be negligible, and then the TDMs dispersions will exhibit a visible mini gap as shown in Fig. R2.

Q2: Does there exist topological phase transition between strong and weak tilted interlayer couplings with the different beta?

Reply: We thank the referee for this very interesting question. To address this question, we have calculated the dispersions of TDMs with strong or weak tilted interlayer couplings at $\beta = 15^\circ$ as an example. The responses of tilted interlayer couplings with $\beta = 26.7^\circ$ are similar. The strength of the tilted interlayer coupling, either strong or weak, can be tuned by changing the parameter w_2 (see in supplementary Fig. 7) of the tilted interlayer coupler. With the decreasing of w_2 , the strength of the tilted interlayer coupling will be weaker as shown in Fig. R2a. Figures R2b-R2e show that TDMs preserve in the band gap until the coupling strength of the tilted interlayer coupler becomes weaker than a critical value, i.e., $I_2/I_1 < 0.5$ ($\theta = \arcsin(\sqrt{I_2/I_1}) < 0.25\pi$); the states (field distribution of a typical one shown in Fig. R2f) bounded at the tilted interlayer couplers region emerge inside the nontrivial band gap as we decrease the coupling strength (decrease w_2). Thus, there is no topological phase transition between strong and weak tilted interlayer couplings.

Figure R2: **a**, Power transmission spectra that characterize the coupling strength induced by the tilted interlayer coupler with different w_2 s. The bulk band gap (ranging from 6.16 kHz to 6.32 kHz) is highlighted in yellow. **b-e**, TDMs dispersions (red and blue dots) for different tilted interlayer couplings realized by changing w_2 of the tilted interlayer coupler for **(b)** $w_2 = 14.7 \text{ mm}$, **(c)** $w_2 = 12 \text{ mm}$, **(d)** $w_2 = 9.5 \text{ mm}$ and **(e)** $w_2 = 5.5 \text{ mm}$. The red and blue dots represent the TDMs for the pseudospin-up and pseudospin-down modes, respectively. The black dots represent the bulk states and the gray dots represent the states bound at the tilted interlayer couplers region. The mini gaps of the TDMs dispersions are due to the pseudospin flip at connections. The system considered in **b** is the same as that in Fig. 2d, where we can see a clear nontrivial band gap and the presence of TDMs inside this gap. As we decrease the parameter w_2 to 5.5 mm, as shown in **a** and **e**, the strength of the tilted interlayer coupling is weak, i.e., $I_2/I_1 < 0.5$ ($\theta = \arcsin(\sqrt{I_2/I_1}) < 0.25\pi$), for the whole bulk gap range and the TDMs disappear. **f**, The pressure amplitude distribution of a state ($f = 6.29 \text{ kHz}$, $k_z = 0$, as marked by the pink dot in **e**) bounded at the tilted interlayer couplers region.

4. It would be better if the authors can compare their 1D topological dislocation states to the other 1D topological hinge states in 3D topological crystals.

Reply: We thank the referee for the good suggestion. Compared with topological hinge states propagating along the usually straight hinges of 3D topological crystals, the TDMs can propagate along the dislocations with various shapes created anywhere inside the bulk or on the surface of 3D topological crystals. Meanwhile, the TDMs based waveguides can support multimode transport by increasing the Burgers vector or the topological invariants for our current system. These peculiar properties can lead to new possibilities in information communication and energy transportation.

In the revised manuscript, we have added the following sentence in the revised manuscript (page 11) “Concretely, for example, compared with topological surface states in 2D which exist only on the boundaries or interfaces, and the 1D topological hinge states propagating along the usually straight hinges of 3D topological crystals⁴¹⁻⁴⁶, the TDMs can propagate unidirectionally along the dislocations with various shapes created anywhere inside the bulk or at the surface of 3D topological crystals¹¹⁻¹⁴.” and rewritten the following sentence in the summary (page 11) “The capability of freely unidirectional routing waves in 3D is far beyond that exhibited in the early reported artificial crystal-based waveguides^{32,33} or the newly developed topological edge/surface/hinge states-based topological waveguides³⁴⁻⁴⁶”, and cited a few new references (Refs. [44-46]) for the 1D topological hinge states in 3D topological

crystals (highlighted in yellow).

5. In Fig. 3d and Fig. 4d, the authors estimate the sound dissipation coefficient is $2.1(1/m)$ along the dislocation path. How about the dissipation coefficient in the air at the same frequency?

Reply: This is a very sharp question. Indeed, the sound dissipation in our confined system is much stronger than the loss in the air. According to reference [ISO 9613-1:1993(E) Acoustics — Attenuation of sound during propagation outdoors] or [GB/T 17247. 1 — 2000 eqv ISO 9613-1:1993], for the sounds propagating in air in free space, the sound dissipation coefficient is a function of four main variables: the frequency of sound, and temperature, humidity and pressure of the air. In our experiment, the sample is in the air with temperature 20°C , relative humidity 60% and pressure 101.325 kPa. Accordingly, the sound dissipation coefficient caused by the atmospheric absorption in air in free space is 0.0574dB/m at 6.3 kHz. In Fig. 3d, the sound dissipation coefficient at 6.27 kHz in dislocation path is fitted to be around $2.1(1/m)$ [$21(\text{dB/m})$], which is much larger than that in free air space due to the viscous dissipation in the strongly confined space along the dislocation path. In fact, the sound dissipation of the viscous dissipation in the confined space has been used in the design of sound absorbers (e.g., micro-perforated absorbers), as shown in the previous studies [e.g. J. Acoust. Soc. Am. 104, 2861–2866 (1998); Appl. Phys. Lett. 108, 063502 (2016); Appl. Phys. Lett. 114, 151901 (2019)]. To address the referee’s question, we measured the sound dissipation coefficient in a straight waveguide with the same cross section as the ring-waveguides in our experiments. The measured data is shown in Fig. R3, wherein the sound dissipation coefficient at 6.27 kHz is fitted to be around $1.9(1/m)$. Here the oscillation of field amplitude is originated from the Fabry–Pérot interference due to the in and out ports of the straight waveguide. We can see the sound dissipation coefficient in dislocation path [$2.1(1/m)$] is slightly larger than that in the straight waveguide, which is due to the complex fine structures (more viscous losses due to the boundary layers) along the dislocation path. Similar results have also been observed in reference Refs. 30 , 31 and 42.

Figure R3: Sound intensity (red circles) measured at 6.27 kHz for a sequence of sites

along a straight waveguide. The black line shows an exponential fit, where l is the propagation length measured from the first site.

Reviewer #3

This manuscript presents a theoretical and experimental study of topological defect modes (TDMs) and bulk-dislocation correspondence in weak 3D acoustic topological insulators. The system under study consists of a 3D topological insulator stacked by layers of 2D quantum spin Hall insulators with weak inter-layer couplings. A screw dislocation is added to the stacking direction to adjust the Burgers vectors, and pairs of TDMs with spin-momentum locking are supported in the bulk gap and are spatially localized at the dislocation. The number of TDM pairs is determined by the Burgers vectors, and allows to probe the bulk topology, in this case, the weak topological indices. Experimentally, the authors study the acoustic waves coupled to the TDMs propagating along the dislocation, measure the dispersion and transmittance of the TDMs, and demonstrate the robustness of the TDMs against spin-conversing perturbations.

Overall, the presented results are new and interesting, and the manuscript is well written and concise. The observation of TDMs and bulk-dislocation correspondence in 3D acoustic systems is timely and may stimulate efforts in other classical wave systems. The manuscript should be of interest to the topological physics community. I thus recommend the manuscript to be accepted in Nature Communications. I also have a few comments regarding the details of the manuscript for the authors to consider.

Reply: We thank the referee for the high evaluation of our work and the recommendation for publication in Nature Communications. We also appreciate your careful reading of our manuscript and many valuable suggestions/comments, which have been carefully addressed below.

1. The manuscript currently is focused on weak 3D TIs, and this is worth mentioning in the title and abstract.

Reply: We thank the referee for this nice suggestion. This has been mentioned in some related places in the text. We tend to retain our title for concise.

2. In the examples studied here, the screw dislocations are all embedded within the bulk of the system to allow unambiguous observation of TDMs. I wonder how the

TDMs interact with the QSH surface states, if, say, the dislocation is added at the xz or yz planes.

Reply: We thank the referee for raising this interesting question. To fully address the referee's question, we consider the system shown in the left panel of Fig. R4, with one dislocation at the center and the other one at the xz boundary. The TDM mode along the dislocation at the boundary will interact with the 3D TI surface states. The band structure of this system is shown in the right panel of Fig. R4. Here gray area represents the projections of bulk bands, black lines are the dispersion of the TDMs at the center, and the red and blue lines are the dispersion of helical states along the boundary. We can see that the TDMs at the boundary hybridize with the 3D TI surface states and form nearly flat bands. The flatness of hybridized bands depends on the size the system. Meanwhile, due to the nontrivial topology, the hybridized surface bands still connect the lower and upper bulk bands. One of the hybridized surface state (denoted by the pink sphere) is shown in the left panel of Fig. R4.

Figure R4: Left panel: A semi-infinite size system (periodic along the z direction) with one dislocation at the center and another dislocation at the xz boundary. The color shows the pressure amplitude distribution of the state labeled by the pink dot in the right panel. Right panel: Dislocation-projected band structure. Gray regions are the projections of bulk bands. The red and blue lines represent the dispersions of the TDMs hybridized with the 3D TI surface states for the pseudospin-up and pseudospin-down modes, respectively. The black lines represent the TDMs trapped by the dislocation at the center which is the same as Fig. 2. The tiny gaps (barely seen) at $k_z = 0$ and $k_z = \pi$ are due to the pseudospin flip at connections.

3. The interlayer couplings are designed to be weak in the current results. I wonder how the TDMs depend on the interlayer couplings? How does the result change, if the xy plane also supports surface states, i.e., for a strong 3D TI?

Q1: I wonder how the TDMs depend on the interlayer couplings?

Reply: This is an inspiring question. To address this question, we have calculated the bulk and TDMs dispersions of system with different interlayer couplings and the data are shown in Fig. R5. The different interlayer couplings are realized by changing the radius r_0 of the vertical tubes (purple in Fig. 2) and the interlayer couplings will be stronger as the radius r_0 increases.

From the data shown in Fig. R5, we can get two basic information. First, the omnidirectional band gap of the bulk will be closed as interlayer coupling becomes stronger; second, the TDMs are present until the band gap along the k_z direction closed.

Figure R5: Bulk band (upper row) and TDMs dispersions (red and blue dots in the lower row) for different interlayer couplings realized by changing the radius r_0 of the vertical tubes for **a** $r_0 = 1 \text{ mm}$, **b** $r_0 = 3 \text{ mm}$ and **c** $r_0 = 5.5 \text{ mm}$. The black dots represent the bulk states. The system considered in **a** is the same as that in Fig. 2d, where we can see a clear nontrivial band gap (yellow) and the presence of TDMs inside this gap. As we increase r_0 to 3 mm , the nontrivial bulk gap almost closes while the TDMs still exist as shown in **b**. When we further increase r_0 to 5.5 mm as shown in **c**, the TDMs disappear.

Q2: How does the result change, if the xy plane also supports surface states, i.e., for a strong 3D TI?

Reply: This is a very interesting question. According to Refs. [11-14], for a 3D strong

topological insulator (TI) characterized by $v_0 \neq 0$, \mathbf{M}_v ($2\mathbf{M}_v = \mathbf{G}_v = \sum_{i=1}^3 v_i \mathbf{b}_i$), the protected topological dislocation modes exist when the product $\mathbf{M}_v \cdot \mathbf{B} \pmod{2\pi}$ is nontrivial. Meanwhile, such a strong TI also supports surface states for any truncated surface, e.g., the xy plane. In such a case, the surface states will hybridize with the TDMs as demonstrate in Ref. [19] in the main text.

Without surface states on the xy surface, our system facilitates the probing of the dislocation states. If there are surface states in the xy plane, TDMs will couple with the surface states, which could lead to fascinating phenomena. As far as we know, the strong 3D TI has not yet been realized in any acoustic system, which remains a challenge to overcome.

4. The authors claim that the topological dislocation transport provides unprecedented control over wave propagations and new possibilities for applications, but do not expand further on the application side. It would be helpful if these claims can be expanded. And comparisons on topological surface states in 2D and TDMs in 3D could also be helpful, since unidirectional wave routing is also achievable in 2D systems.

Reply: We thank the referee for this constructive suggestion. Accordingly, we have added the following sentence in the revised manuscript (page 11) “Concretely, for example, compared with topological surface states in 2D which exist only on the boundaries or interfaces, and the 1D topological hinge states propagating along the usually straight hinges of 3D topological crystals⁴¹⁻⁴⁶, the TDMs can propagate unidirectionally along the dislocations with various shapes created anywhere inside the bulk or at the surface of 3D topological crystals¹¹⁻¹⁴.” and rewritten the following sentence in the revised manuscript (page 11) “The capability of freely routing waves in 3D is far beyond that exhibited in the early reported artificial crystal-based waveguides^{32,33} or the newly developed topological edge/surface/hinge states-based topological waveguides³⁴⁻⁴⁶.” and corresponding references have been added (high light with yellow in Refs. [42-46]). In original manuscript, we also discuss the application of TDMs (pages 6-7) “The exceptional performance of sound routing in 3D space will be particularly useful for designing new conceptional acoustic devices.” and (page 11) “On the application side, the peculiar topological dislocation transport points to new possibilities for information communication and energy transportation.”

REVIEWER COMMENTS

Reviewer #1 (Remarks to the Author):

The authors have made much revisions to the main text and supplementary information. Nevertheless, I think the absence of a complete theory will weaken the paper. Even though a Hamiltonian of the acoustic "weak topological insulator" cannot be explicitly constructed, there are other ways to strengthen the authors' claim of weak topological insulator and topological invariants. The edge states, which are used in the revised SI to support the conclusion that the system has a topological band gap, are not sufficient proof topological band gap. There are many cases that the edge states can be tuned away. Therefore, I suggest the authors to make a more sound theory in the SI to establish that the band gap mimic a quantum spin Hall insulator. There are several possibilities for this purpose: the symmetry indicators or the Floquet topological insulator theory, or Berry phase calculations. Only when this theory is well established, the authors claim of the whole story can be stand on a firm ground.

Reviewer #2 (Remarks to the Author):

In the revised version, all the issues arising in the first round of review have been well addressed. I have no further comments. Now, I am happy to recommend this manuscript for publication in Nature Communications.

Reviewer #3 (Remarks to the Author):

In the response letter and the revised manuscript, the authors have addressed my comments adequately and revised the manuscript to incorporate my suggestions. I believe that the manuscript should be publishable in Nature Communications without further revisions.

Reviewer #1

The authors have made much revisions to the main text and supplementary information.

Reply: We thank the referee for the great efforts in reviewing our revised manuscript.

Nevertheless, I think the absence of a complete theory will weaken the paper. Even though a Hamiltonian of the acoustic "weak topological insulator" cannot be explicitly constructed, there are other ways to strengthen the authors' claim of weak topological insulator and topological invariants. The edge states, which are used in the revised SI to support the conclusion that the system has a topological band gap, are not sufficient proof topological band gap. There are many cases that the edge states can be tuned away. Therefore, I suggest the authors to make a more sound theory in the SI to establish that the band gap mimic a quantum spin Hall insulator. There are several possibilities for this purpose: the symmetry indicators or the Floquet topological insulator theory, or Berry phase calculations. Only when this theory is well established, the authors claim of the whole story can be stand on a firm ground.

Reply: We thank the referee for raising this point. To respond to this point, we have added a new section in supplementary information (pages 7-8) to outline the basic results of the transfer matrix method for the 3D weak insulator stacked by layers of 2D quantum spin Hall insulators with weak interlayer coupling. We have also revised the corresponding sentences in the main text to address this point (page 5) "The vertical tubes are printed narrow for fulfilling weak interlayer coupling, such that the nontrivial topology of the 2D QSH system is inherited (see supplementary Fig.5), meanwhile, with the careful design of their distribution, to avoid visible pseudospin flip (see supplementary Fig. 6)."

Reviewer #2

In the revised version, all the issues arising in the first round of review have been well addressed. I have no further comments. Now, I am happy to recommend this manuscript for publication in Nature Communications.

Reply: We thank the referee for the kind recommendation for publication in Nature Communications.

Reviewer #3

In the response letter and the revised manuscript, the authors have addressed my comments adequately and revised the manuscript to incorporate my suggestions. I believe that the manuscript should be publishable in Nature Communications without further revisions.

Reply: We thank the referee for the kind recommendation for publication in Nature Communications.

REVIEWER COMMENTS

Reviewer #1 (Remarks to the Author):

After carefully read the revised manuscript and the reply, I have to come to the conclusion that the authors have not done their job satisfactorily. Clearly, the revised supplementary information is not adequate for the establishment of the topological invariants of the "acoustic Floquet topological insulator". I can understand that the authors are using the winding number to indicate the band topology. However, this part in the supplementary is made too brief, even I can hardly find any reasoning that could connect the winding numbers to the band topology and eventually to the claimed $(0,0,1)$ index of the weak topological insulator in the main text. Therefore, I still see there is apparently a gap that the authors must fill to establish there conclusions firmly.

Dear Dr. Cristiano Matricardi,

Thank you very much for communicating the manuscript (NCOMMS-21-13865B). We also appreciate all the referees for their comments and suggestions.

In the last round of reports, two of the referees supported our work for publication, while the first referee was not satisfied with the establishment of the topological invariants for our system. He/she thought that part was “made too brief” and the connection between “winding numbers to the band topology...” was not clear. To address the concerns raised by the first referee, we add more details for that, together with a physical interpretation, in Supplementary Fig. 5, as highlighted in yellow

We think our current version should be suitable for publication in Nature Communications. Thank you very much for your reconsideration.

Yours sincerely,

Chunyin Qiu, Meng Xiao, Manzhu Ke and Zhengyou Liu

Summary of changes

1. The section of Supplementary Fig. 5 has been rewritten in the Supplementary Information and the corresponding references have been added, as highlighted in yellow. Ref. [8] of the original version has been relabeled as Ref. [9] accordingly.

Reviewer #1 (Remarks to the Author):

After carefully read the revised manuscript and the reply, I have to come to the conclusion that the authors have not done their job satisfactorily. Clearly, the revised supplementary information is not adequate for the establishment of the topological invariants of the "acoustic Floquet topological insulator". I can understand that the authors are using the winding number to indicate the band topology. However, this part in the supplementary is made too brief, even I can hardly find any reasoning that could connect the winding numbers to the band topology and eventually to the claimed (0,0,1) index of the weak topological insulator in the main text. Therefore, I still see there is apparently a gap that the authors must fill to establish there conclusions firmly.

Reply: We thank the referee for raising this point. As demonstrated in Refs. [28-31], the anomalous Floquet insulators (AFIs) support chiral edge modes in the band gaps between bulk bands of zero Chern numbers. The topological properties of AFIs can be characterized by the topological invariants (winding numbers) ν associated with the bulk evolution operator [see Phys. Rev. X 3, 031005 (2013)] or topological boundary invariants n [see Phys. Rev. B 89, 075113 (2014)]. Our two-dimensional (2D) acoustic AFI constructed by 2D strong coupled waveguides network, which shares the same physics as Refs. [29-31]. For these network models, since the Hamiltonian of the system cannot be written explicitly, the exact bulk evolution operator cannot be given, so that the topological invariants ν cannot be calculated. Based on an adiabatic pumping picture for the AFIs constructed with strong coupled waveguides network, YD Chong et al. proposed an alternative topological invariant, i.e., topological boundary invariant n , to characterize such systems [Phys. Rev. B 89, 075113 (2014)], which is adopted in our work to characterize the topological feature of our 3D system with fixed k_z (effectively two-dimensional). If the topological invariant n does not change with k_z , we can conclude that our system is a weak topological insulator. This is evidenced in our work, as the result of the weak interlayer coupling.

The adiabatic pumping picture for characterizing the topological systems was originally introduced by Laughlin [Phys. Rev. B 23, 5632 (1981)] and has been widely used in both the time dependent and static systems [e.g., Phys. Rev. B 84,195410 (2011); Phys. Rev. B 89, 075113 (2014)]. Similar to 2D topological

insulators, topologically nontrivial pumps are characterized by the appearance of gapless edge states during the course of a pumping cycle. YD Chong et al. demonstrated that a related procedure can be carried out for the strongly coupled network model discussed herein. They confirmed that the edge states of the network system can be counted by an adiabatic pumping invariant (topological invariant n) based on the winding number of the coefficient of reflection at one edge of the network. More details can be found in reference [Phys. Rev. B 89, 075113 (2014)] and the newly added discussions in Supplementary Information (pages 7-9).

Our 3D system is constructed by layers of 2D strong coupled waveguide networks and thus we face the same difficulty as the corresponding 2D system discussed by Chong et al [Phys. Rev. B 89, 075113 (2014)] – we cannot give the explicit expression for the bulk evolution operator. Thus, we follow Chong et al's work and calculate the adiabatic pumping based topological boundary invariant n to characterize the topological properties of our 3D system. We have added a detailed discussion of the winding number (topological boundary invariant n) for our 3D system in the revised supplementary information (pages 7-9).

REVIEWERS' COMMENTS

Reviewer #1 (Remarks to the Author):

The authors have satisfactorily addressed the issue of topological indices in the supplementary information with sufficient details. I fully recommend the publication of this paper.